# Process Optimization, Microstructure and Mechanical Properties of Wire Arc Additive Manufacturing of Aluminum Alloy by Using DP-GMAW Based on Response Surface Method

**DOI:** 10.3390/ma16165716

**Published:** 2023-08-21

**Authors:** Wenbo Du, Guorui Sun, Yue Li, Chao Chen

**Affiliations:** 1National Key Laboratory for Remanufacturing, Army Academy of Armored Forces, Beijing 100072, China; 2College of Mechanical and Electrical Engineering, Northeast Forestry University, Harbin 150040, China; 3Beijing Institute of Radio Measurement, Beijing 100039, China

**Keywords:** double-pulsed gas metal arc welding, additive manufacturing, response surface method, parameter optimization, microstructure analysis, aluminum alloy

## Abstract

Double-pulsed gas metal arc welding (DP-GMAW) is a high-performance welding method with low porosity and high frequency. Periodic shrinkage and expansion of the melt pool during DP-GMAW leads to unusual remelting, and the re-solidification behavior of the weld metal can significantly refine the weld structure. The advantages of DP-GMAW have been proven. In order to better apply DP-GMAW to aluminum alloy arc additive manufacturing, in this paper, the single-pass deposition layer parameters (double-pulse amplitude, double-pulse frequency and travel speed) of DP-GMAW will be optimized using the response surface method (RSM) with the width, height, and penetration of the deposition layer as the response values to find the superior process parameters applicable to the additive manufacturing of aluminum alloy DP-GMAW. The results show that the aluminum alloy components made by DP-GMAW additive are well formed. Due to the stirring of double-pulse arc and the abnormal remelting and solidification of metal, the microstructures in the middle and top areas show disordered growth. The average ultimate tensile strength of the transverse tensile specimen of the member can reach 175.2 MPa, and the elongation is 10.355%.

## 1. Introduction

Aluminum material is one of the important materials for lightweight design. Aluminum products have a large number of applications in the fields of aerospace, vehicles, and mechanical preparation [1,2,3]. Wire and Arc Additive Manufacturing (WAAM) is a typical process of Additive Manufacturing (AM) technology by digital means. Compared to traditional aluminum alloy manufacturing methods, such as casting, forging, and welding, WAAM has many advantages in terms of simplicity, freedom of design, and high material utilization [4,5,6,7,8].

Double-pulsed gas metal arc welding (DP-GMAW) is a high-performance welding method with low porosity, high frequency, and concentrated energy [9,10]. Numerous scholars have shown that the microstructure of DP-GMAW welds is significantly finer compared to pulsed gas metal arc welding (P-GMAW). The shear force at the peak of the strong pulse causes the dendrites to break up, providing enough nuclei for grain growth [11,12,13]. Based on the advantages of DP-GMAW, some scholars have studied DP-GMAW additive manufacturing and verified the feasibility of DP-GMAW additive manufacturing. Mainak Sen et al. [11] conducted overlay tests on mild steel plates using DP-GMAW with different combinations of parameters. The results show that the volume fraction of inclusions and needle ferrite in the weld metal increases with decreasing heat input, pulse frequency, and thermal pulse frequency. Yao P et al. [13] showed that the width, height, and depth of melt of the weld seam were positively correlated with the double-pulse relationship, average welding current, and percentage change in double-pulse current, and negatively correlated with travel speed and double-pulse frequency. Koushki A R et al. [14] added 0.1 vol% of oxygen and nitrogen to the shielding gas and the tensile and flexural strength of the weld was improved; however, the formation of oxide film was detrimental to the performance when the content was further increased.

The response surface method (RSM) is to fit the equation from the data of the experiment and represent it by means of a coordinate plot, which can predict the effect of different conditions on the response values. Moreover, the results can be optimized under specific conditions [15,16]. Haibin Geng et al. [17] developed a predictive model between the input variables (peak current, wire feed speed, and travel speed) and the response values (height and width of weld bead). The validity of the model was tested by analysis of variance (ANOVA). Waheed et al. [18] optimized the welding process parameters by RSM. Using optimal welding parameters optimizes the welding sequence. The results showed a 19% reduction in overall deformation caused by welding. Karganroudi et al. [19] used RSM to analyze the effect of current size and welding speed on weld geometry and temperature distribution. The penetration depth and penetration width decreased with the reduction in current residence time. Youheng F et al. [20] optimized the bainitic steel additive manufacturing process parameters using the RSM. The optimized specimen has a smooth surface with less splash and no visible defects. Escribano-García R et al. [21] combined RSM and finite element method (FEM) for 3D numerical simulation of GMAW cold metal transfer. They used RSM to find the optimal parameters. Sarathchandra D et al. [22] studied the effect of process parameters on WAAM of 304 stainless steel. RSM and ANOVA were used to evaluate the effects of current, travel speed, and weld distance on weld seam characteristics.

The advantages of DP-GMAW have been proven. In order to better apply DP-GMAW to aluminum alloy arc additive manufacturing, in this paper, the RSM was used to optimize the parameters of DP-GMAW additive manufacturing process, the effects of DP-GMAW process parameters on each response quantity of single-layer single-pass deposition layer of aluminum alloy were compared and analyzed, and a mathematical model was established between three process parameters of travel speed, double-pulse frequency, and double-pulse amplitude, and three response values of deposition layer width, height, and penetration. The regression model ANOVA of the width, height, and penetration of the deposition layer was also performed to check the significance of the model and the normal probability distribution of the model residuals. The relationship between the variables (travel speed, double-pulse frequency, double-pulse amplitude) and the response values (width, height, and penetration of the deposition layer) was analyzed using perturbation plots. A set of optimized parameters was selected for the deposition and forming of thin-walled components of ER4047 aluminum alloy, and their microstructure and mechanical properties were analyzed.

## 2. Materials and Methods

The Fast Mig X 350 (KEMPPI, Lahti, Finland) welder was selected for the test, and the current waveform was selected as a double-pulse waveform. The test system is shown in Figure 1a. The waveforms of typical welding currents during DP-GMAW [13] are shown in Figure 1b. The current waveform is a double-pulse cycle consisting of a strong pulse group (Pulse S) and a weak pulse group (Pulse W). Test material selection of 1.2 mm ER4047 aluminum alloy wire, substrate selection to remove the oxide layer of 2A12 aluminum alloy substrate, and wire and substrate composition are shown in Table 1. The protective gas (99.99% pure argon) flow rate was selected at 20 L/min, and the wire feed speed was fixed at 6 m/min. The traveling speed was precisely controlled by the single axis motion control box, and the speed value was measured and feedback by the sensor was installed on the electric ball screw system.

Box–Behnken design tests were selected using Design-Expert 10.0.7 software of StatEase company (Minneapolis, MN, America). Three key process parameters that control the morphology of the DP-GMAW additive deposition layer were selected as the study variables for the experiment, i.e., double-pulse frequency (F), double-pulse amplitude (A), and travel speed (V). The three key parameters of deposition layer width, height, and penetration were selected as response values. In order to investigate the fitting of the central area and ensure the repeatability of the test, the central point repeated test is set to five groups. In order to improve the arc additive forming rate, the range of process parameters was determined as shown in Table 2, based on a large number of preliminary experimental explorations, combined with the forming quality of the single-pass deposition layer.

The RSM test parameters are shown in Table 3. Metallographic specimens were cut at the highest and lowest positions using a CNC wire cutter. After grinding with sandpaper and polishing with a metallographic polisher, they were etched using Keller’s reagent. The cross-sectional morphology was photographed using a body microscope, and the values of single-pass deposition layer width, height, and fusion depth were measured separately. Since some of the parameters are fluctuations in the formation of the deposition layer surface, the average value of the two cross-sections was taken as the response quantity. Figure 2 shows the surface morphology of the deposition layer. The surface morphology of the deposition layer under different parameters is significantly different, as shown in Figure 2 (1–17).

## 3. Results and Discussion

ANOVA was used to test the significance of the model and its misfit. The feasibility of the model is judged based on the magnitude of the Adeq Precision value. The functional relationship between the input variables and the response quantity can be expressed uniformly as y = f(*A*, *F*, V), expanding the response quantity y into the form of a second-order polynomial regression equation. After the calculation of the obtained coefficients, the functions between the deposition layer width, height, and penetration and the three input variables are shown in Equations (1)–(3), respectively. By analyzing the regression equation and combining the test results of the response in Table 3, the mathematical relationship between the input variables and the response within the parameter range is established to seek the optimal process parameters. The coefficients in the equation have been retained to three decimal places.
(1)$$\begin{array}{l} P=+1.57+0.051\cdot A-0.074\cdot B-0.084\cdot C+0.027\cdot AB+0.081\cdot AC+0.11\cdot BC\\ +0.024\cdot A^{2}+0.1\cdot C^{2}+0.11\cdot A^{2}B \end{array}$$
(2)$$\begin{array}{l} W=+6.39+0.27\cdot A+0.069\cdot B-0.00875\cdot C-0.28\cdot AB+0.33\cdot BC\\ +0.086\cdot A^{2}-0.27\cdot B^{2}+0.26\cdot C^{2}-0.52\cdot A^{2}B \end{array}$$
(3)$$\begin{array}{l} H=+1.91-0.059\cdot A-0.072\cdot B-0.12\cdot C-0.014\cdot AB+0.17\cdot AC+0.18\cdot BC\\ +0.20\cdot A^{2}+0.040\cdot B^{2}-0.23\cdot C^{2} \end{array}$$

From Table 4, it can be seen that the model *F* value for penetration is 5.21. The value of probability *P* > *F* is less than 0.05, indicating that the model is significant and the value of probability *P* > *F* is 0.0203 (less than 0.05), so the model is significant and statistically significant. The coefficient of determination R-Squared(R^2^) value of the fitted regression equation is 0.8701 (>0.80), which is relatively close to 1. These two points indicate that the fitted equation of the model is acceptable. The value of Lack of Fit (LF) is 1.22, which results in insignificant, indicating that the model is reliable. In determining the penetration depth, the influence of *C* travel speed is important, while *A* and *B* are not significant in determining the penetration depth. A signal-to-noise ratio greater than 4 is ideal, and the model’s signal-to-noise ratio of 8.625 indicates that there is sufficient signal for the model to be used for prediction. According to the test results, when *C* value increases from 10 mm/s to 14 mm/s, the value of penetration *P* decreases gradually. This is due to the fact that the increase in travel speed will lead to a reduction in heat input per unit length and the reduction in the depth of molten substrate in the single factor change.

As seen in Table 5, the model *F* value for width is 8.85, indicating that the model is significant. The value of LF is 0.34, which indicates that the model is reliable. Due to noise, there is a 79.90% probability that such a large “Lack of Fit *F*-value” will occur. The R^2^ value of the coefficient of determination of the fitted regression equation is 0.9192 (greater than 0.80), which is closer to 1. A double-pulse amplitude has a significant impact on the width, while B double-pulse frequency and the C travel speed have no significant impact on the width. The signal-to-noise ratio of the model is 10.302, which indicates that the model has sufficient signal to be used for prediction. When A value rises from 0.3 m/min to 1.7 m/min, the overall heat input increases, resulting in a significant increase in the melting width of the substrate.

The model *F*-value for height H is 4.25, which is significant as shown in Table 6. A “misfit *F*-value” of 0.49 means that the model is reliable. There is a 70.57% probability that such a large “out-of-fit *F*-value” will occur due to noise. The R^2^ value of the coefficient of determination of the fitted regression equation is 0.8454 (greater than 0.80), which is closer to 1. The signal-to-noise ratio of the model is 6.952, indicating that there is sufficient signal for the model to be used for prediction. When the traveling speed is constant, the height of the deposition layer is mainly affected by the wire feeding speed. It can be seen from Table 6 that the influence of the amplitude of double pulse on the height of the deposition layer is not significant.

Figure 3a–c shows the normal probability distributions of the residuals of the single-pass deposition layer width, height, and penetration models. The colors in Figure 3 represent responder values of different sizes. The residual distributions of all three models are approximately along a straight line, indicating that the regression models fit better, the error distribution is more uniform, there are no singularities with large deviations, and the models can predict the response quantity values more accurately. Understanding the influence of each input variable (A, B, C) and its interaction terms (such as AB, AC, BC, etc.) on the nature of the response quantity can predict more accurately the trend of the deposition layer size with the change of the double-pulse process parameters.

In order to deeply analyze the effects of the three input variables of double-pulse frequency (F), double-pulse amplitude (A) and travel speed (V) and their interaction terms on the response values, the perturbation plots of the deposition layer width (W), height (H), and penetration (P) were carefully studied, as shown in Figure 4a–c. From Figure 4a, it can be seen that travel speed has the largest amount of perturbation and the most significant effect on the penetration. When the value of the double-pulse amplitude deviates from the central reference point, the penetration of the deposition layer tends to increase gradually as travel speed decreases. This is due to the fact that the smaller the travel speed is, the greater the heat input per unit length and the greater the melting depth of the base material per unit area, i.e., the greater the melting depth. The width of the deposition layer is most significantly influenced by the double-pulse amplitude, as shown in Figure 4b. With the increase in double-pulse amplitude, the width increases significantly. In the DP-GMAW process, the double-pulse amplitude is regulated by the wire feed speed and increasing the double-pulse amplitude means increasing the wire feed speed in the pulse W phase. Therefore, width increases with the increase in double-pulse amplitude. The effects of double-pulse amplitude and travel speed on the deposition layer height are most obvious, and the effects of both on the deposition layer height show opposite trends, as shown in Figure 4c. The deposition layer height tends to decrease and then increase with the increase in double-pulse amplitude, and then increase and then decrease with the increase in travel speed. The deposition layer height is related to the ratio of WFS and travel speed [17] and is influenced by the interaction between the two.

After analyzing and verifying the reliability of the model, the DP-GMAW process parameters are optimized. In the DP-GMAW enrichment process, the double-pulse amplitude agitates the molten pool by modulating the pulse behavior, and the double-pulse frequency mainly controls the transition between the strong and weak pulse groups. The greater the frequency of the double pulse, the more frequent the transition between the strong and weak pulse groups, and the stronger the stirring effect on the molten pool. The grain size can be refined by the stirring effect of the electric arc [13,23]. Therefore, in the “Criteria” tab, set the double-pulse frequency f to Goal; maximize and the double-pulse amplitude to Goal; minimize, and select the scanning speed in the range of 10–14 mm/s as “in range”. The feasibility index distribution of the optimized solution is obtained as shown in Figure 5, and the optimal parameters are obtained near the red area in the figure, and the feasible value of the solution is 1. The optimized results of this solution are double-pulse amplitude of 0.3 m/min, double-pulse frequency of 7 Hz, and travel speed of 12 mm/s.

To investigate the effect of optimized parameters on DP-GMAW additive manufacturing, thin-walled Al-Si alloy parts with a height of 45 mm were prepared using the optimized parameters. The metallographic were cut at the positions shown in Figure 6b. The material is well connected between layers and no macroscopic defects, such as visible holes, cracks, and unfused layers, are found during or after deposition. The stirring action of the double pulse increases the melt pool width and reduces the layer height, measuring an effective wall width of about 8.6 mm. Figure 7 shows the microstructure of the specimen. There are differences between the interlayer and intra-layer microstructures, and the reason for the differences is that the interlayer microstructure has undergone remelting. The grains in the layer grow rapidly along the direction of maximum temperature gradient. The grains within the layer are mainly columnar crystals growing perpendicular to the fusion line, as shown in Figure 7c. Due to the stirring effect of the double pulses, under the influence of alternating periodic arc force, the columnar grains at some locations are interrupted, showing disordered growth of shorter columnar grains, as in Figure 7a,b.

There is less eutectic content between the layers. Due to the partial remelting effect of the arc on the deposition layer, part of the eutectic Si agglomerates in the molten pool are completely melted, and the other part adheres to the bottom of the pool to form large-size eutectic Si agglomerates, or the unmelted eutectic Si continues to grow. The eutectic Si in the interlayer microstructure is larger and sparser compared to the intra-layer size, as shown in Figure 7f. Figure 8 shows the EDS results, α-Al grains are mainly dendritic formations. Most silicon elements are concentrated in Al-Si eutectic, as shown in Figure 8d.

The Al elements form α-Al dendrites except for the eutectic organization with Si elements. The α-Al dendrites in the bottom layer grow approximately parallel and long, while the α-Al dendrites in the middle and top regions grow in a disordered crossed state. In addition to the stirring effect of the double-pulse arc on the melt pool, the grain growth is also related to the heat dissipation of the melt pool. When the initial layer is deposited, the heat from the melt pool can be easily directed to the substrate for heat dissipation, and a small portion of the heat is dissipated through the air by convection and radiation, and the dendrites grow rapidly in the direction of the temperature gradient. When the top layer is deposited, most of the heat can only be radiated through the air by convection and radiation due to the large amount of heat accumulated in the previous deposition layers. Therefore, the growth pattern of α-Al dendrites at the top of the incremental body shows interlocking growth. When the middle layer is deposited, the temperature of the incremental body and the environment are relatively stable, the temperature gradient of the melt pool is small, and the α-Al dendrites grow along the vertical direction of the fusion line. Because of the stirring effect of the double-pulse arc, the α-Al dendrite growth is interrupted and shows a staggered growth pattern.

Figure 9 shows the drawing of tensile results, the sampling position of tensile specimens in transverse and vertical directions, the dimensions of tensile specimens, as shown in Figure 9a,b, and the ultimate tensile strength and elongation, as shown in Figure 9c. The results show that the ultimate tensile strength and elongation of the transverse tensile specimens are higher than those of the vertical ones. The average ultimate tensile strength of the transverse tensile specimens was 175.2 MPa and the average elongation was 10.355%. The average ultimate tensile strength of the vertical tensile specimens was 154.25 MPa and the average elongation was 7.46%. The mechanical properties exhibited significant anisotropy.

Figure 10 shows the SEM of the fracture of the tensile specimens. Air pores and a large number of dimples were found in both transverse and vertical tensile specimen fractograms. Dimples are a typical feature of ductile fracture. During the WAAM process, due to the high solubility in liquid aluminum and the high thermal conductivity and fast cooling of aluminum alloy, the hydrogen in the melt pool cannot escape in time and exists in the form of gas in the melt pool to form pores. The pores over 50 μm will have an impact on the mechanical properties of the components.

## 4. Conclusions

In order to better apply DP-GMAW to aluminum alloy arc additive manufacturing, the impact of three important parameters of DP-GMAW on the three evaluation indicators was analyzed by response surface method, and the following conclusions were drawn:The models between the variables (travel speed, double-pulse frequency, and amplitude) and the response values (width, height, and fusion depth of the deposition layer) were significant. The models all had signal-to-noise ratios greater than 4, with adequate signal.The residual distributions of all three models were approximately along a straight line and the regression models fitted well. The perturbation diagram shows that the penetration is most strongly perturbed by the travel speed, with a gradual increase in penetration as the travel speed decreases. The width of the layer is most significantly influenced by the double-pulse amplitude, while the height of the layer is influenced by the interaction between the double-pulse amplitude and the travel speed.No macroscopic defects were observed during or after the deposition. The thin-walled parts are well formed with an effective wall width of 8.6 mm. Because of the stirring effect of the double-pulse arc, the growth of some α-Al dendrites within the layers is interrupted and shows a staggered growth pattern. The interlayer microstructure undergoes remelting and differs from the intra-layer microstructure.The mechanical properties show a clear anisotropy. The average ultimate tensile strength of the transverse tensile specimens was 175.2 MPa and the average elongation was 10.355%. The average ultimate tensile strength of the vertical tensile specimens was 154.25 MPa and the average elongation was 7.46%.

## Figures and Tables

**Figure 1 materials-16-05716-f001:**
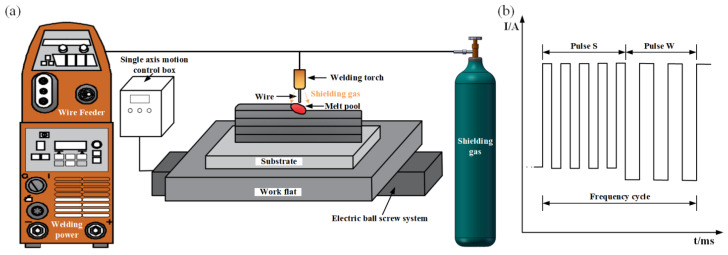
Diagram of the test system: (**a**) diagram of the Wire and Arc Additive Manufacturing (WAAM) process; (**b**) typical double pulse waveform diagram.

**Figure 2 materials-16-05716-f002:**
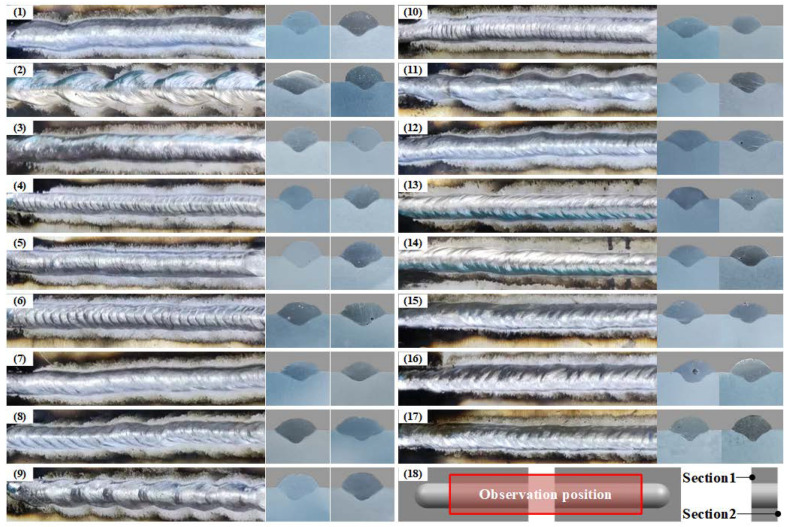
Macroscopic diagram of the deposition layer: (**1**–**17**) surface morphology and cross-sectional view of the deposition layer; (**18**) schematic diagram of sampling and shooting locations.

**Figure 3 materials-16-05716-f003:**
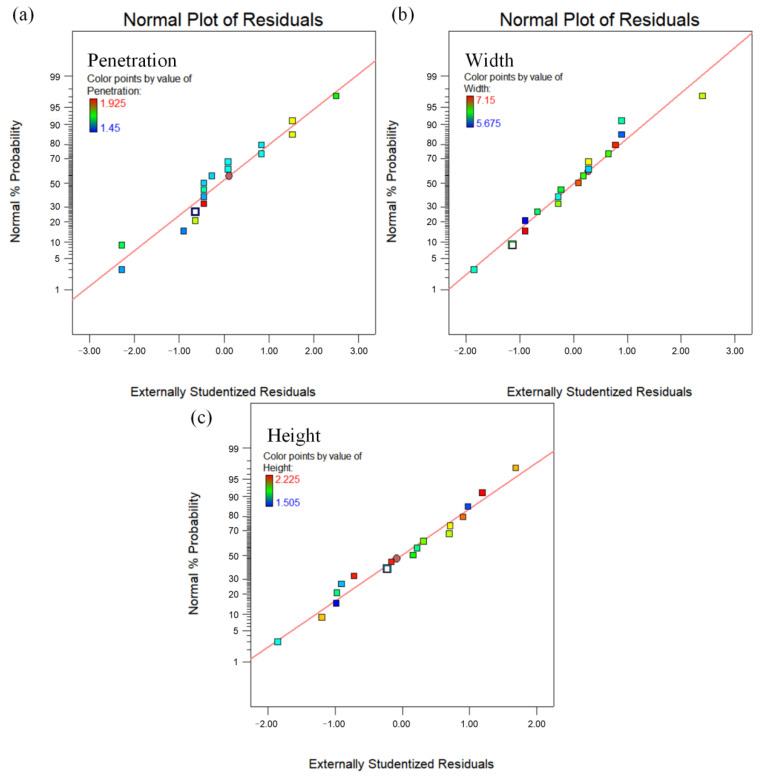
Normal plot of residuals for W, H and P: (**a**) penetration; (**b**) width; (**c**) height.

**Figure 4 materials-16-05716-f004:**
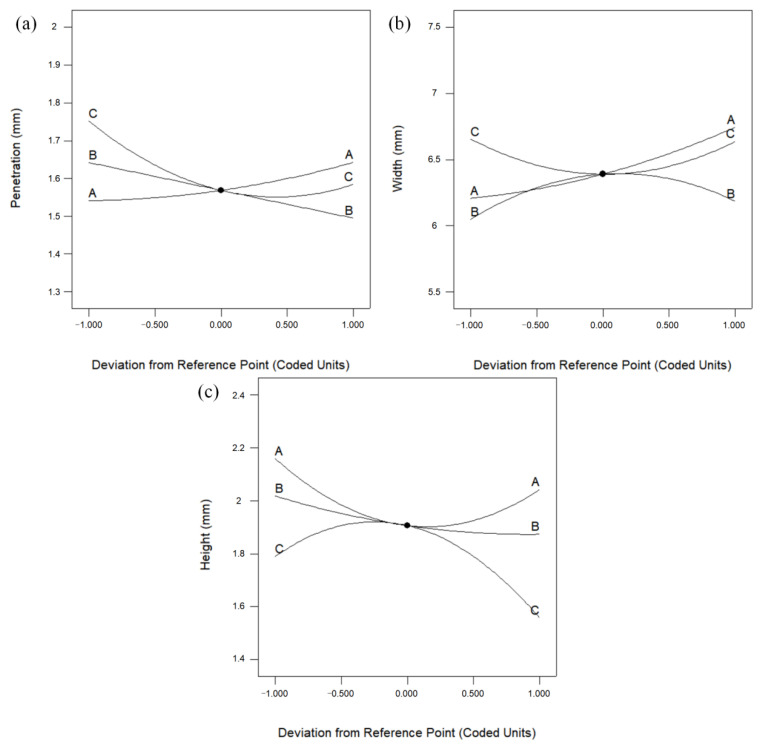
Perturbation diagrams of the deposition layer width, height, and penetration as a function of deviation of center reference point: (**a**) penetration; (**b**) width; (**c**) height. A: Double-Pulse Amplitude; B: Double-Pulse Frequency; C: Travel Speed.

**Figure 5 materials-16-05716-f005:**
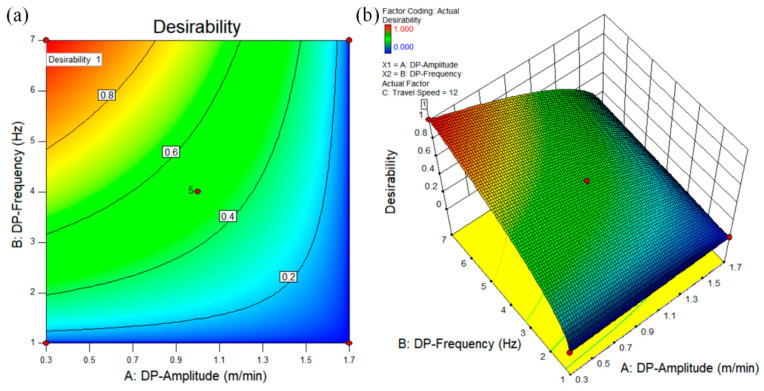
Effect of double-pulse amplitude and double-pulse frequency on the feasibility of the scheme with a travel speed of 12 mm/s: (**a**) contour plot of scheme feasibility; (**b**) 3D surface plot of probability distribution of scheme feasibility.

**Figure 6 materials-16-05716-f006:**
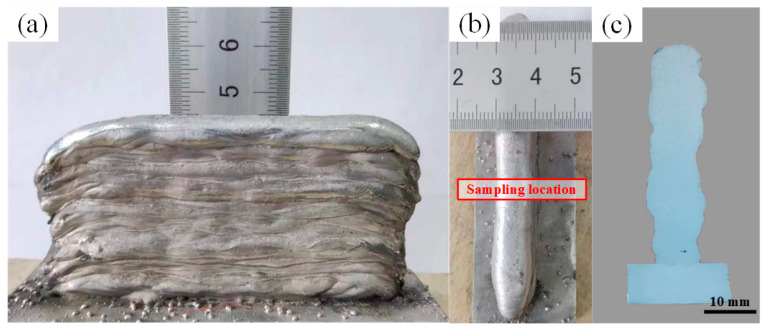
Shape of the WAAM components: (**a**) sidewall morphology; (**b**) wall width; (**c**) metallographic specimen.

**Figure 7 materials-16-05716-f007:**
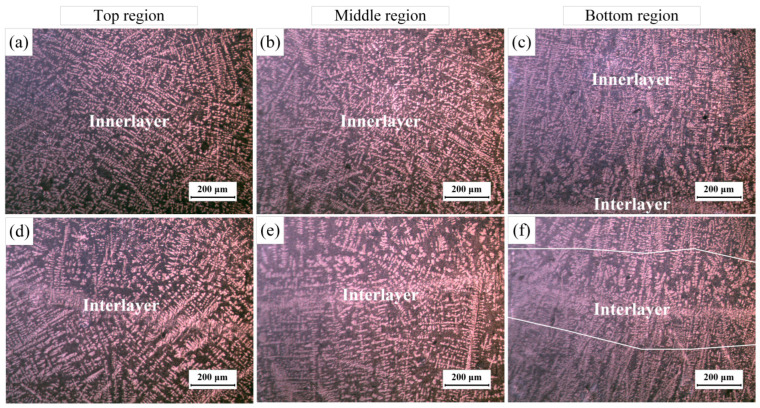
Microstructure at different positions: (**a**–**c**) intra-layer microstructure at the top and middle and bottom positions, respectively; (**d**–**f**) are the interlayer microstructures at the upper and middle positions, respectively.

**Figure 8 materials-16-05716-f008:**
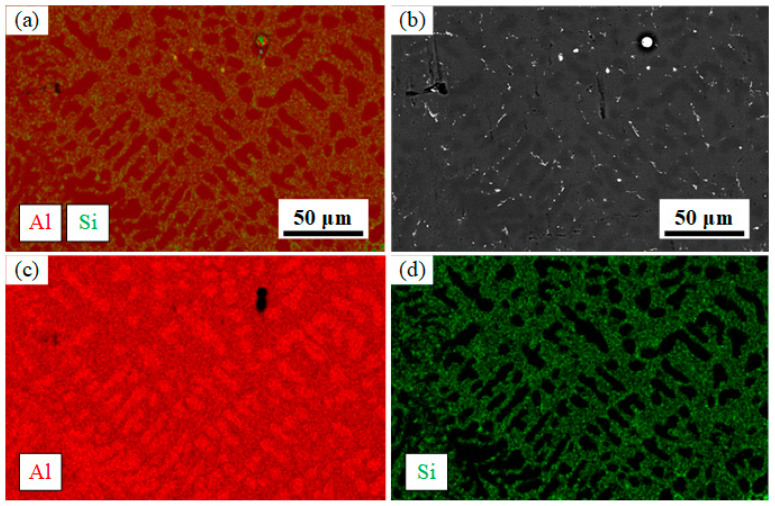
EDS results: (**a**) elemental layering images; (**b**) scanning electron microscope image; (**c**) Al element distribution; (**d**) Si element distribution.

**Figure 9 materials-16-05716-f009:**
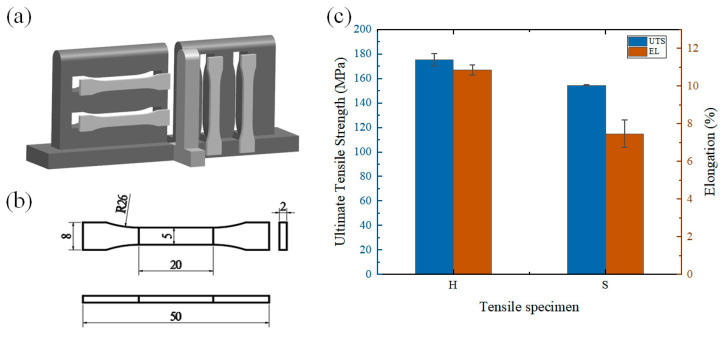
Drawing of tensile specimen: (**a**) sampling position of tensile specimen; (**b**) size of tensile specimen (mm); (**c**) ultimate tensile strength (UTS) and elongation (EL) of specimen. H—horizontal tensile specimen, S—vertical tensile specimen.

**Figure 10 materials-16-05716-f010:**
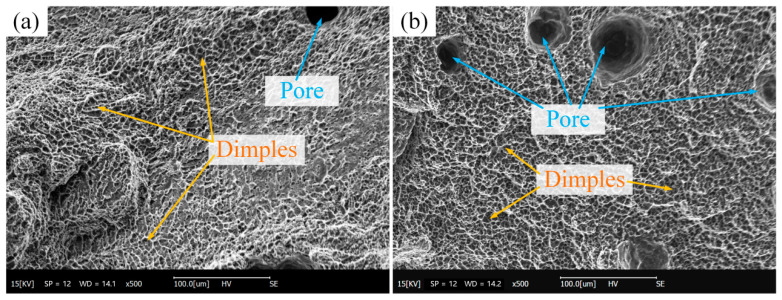
Fracture morphology (500×): (**a**) tensile specimen H; (**b**) tensile specimen S.

**Table 1 materials-16-05716-t001:** Nominal compositions of ER4047 wire and 2A12 substrate (wt.%).

Element	Si	Fe	Cu	Mg	Mn	Ti	Zn	Al
ER4047	11~13	≤0.6	≤0.3	≤0.1	≤0.15	≤0.15	≤0.2	balance
2A12	≤0.50	0~0.5	3.8~4.9	1.2~1.8	0.30~0.9	≤0.15	≤0.30	balance

**Table 2 materials-16-05716-t002:** Input process parameters and working ranges.

Parameters (Unit)	Optimization Scope		
	−1	0	1
A (m/min)	0.3	1	1.7
F (Hz)	1	4	7
V (mm/s)	10	12	14

**Table 3 materials-16-05716-t003:** The response surface method (RSM) test parameters and corresponding response.

Std Order	Design Matrix			Input Variables			Responses		
				A: Double-Pulse Amplitude/(m/min)	B: Double-Pulse Frequency/(Hz)	C: Travel Speed/(mm/s)	Penetration P/(mm)	Width W/(mm)	Height H/(mm)
1	−1.00	−1.00	0.00	0.3	1	12	1.560	6.165	2.205
2	1.00	−1.00	0.00	1.7	1	12	1.525	7.150	2.085
3	−1.00	1.00	0.00	0.3	7	12	1.575	5.825	2.225
4	1.00	1.00	0.00	1.7	7	12	1.650	5.675	2.050
5	−1.00	0.00	−1.00	0.3	4	10	1.780	6.500	2.200
6	1.00	0.00	−1.00	1.7	4	10	1.800	7.025	1.775
7	−1.00	0.00	1.00	0.3	4	14	1.450	6.310	1.625
8	1.00	0.00	1.00	1.7	4	14	1.795	7.100	1.875
9	0.00	−1.00	−1.00	1	1	10	1.925	6.625	2.150
10	0.00	1.00	−1.00	1	7	10	1.550	6.100	1.505
11	0.00	−1.00	1.00	1	1	14	1.530	5.985	1.559
12	0.00	1.00	1.00	1	7	14	1.610	6.785	1.640
13	0.00	0.00	0.00	1	4	12	1.690	6.505	1.700
14	0.00	0.00	0.00	1	4	12	1.510	6.350	1.780
15	0.00	0.00	0.00	1	4	12	1.575	6.275	2.100
16	0.00	0.00	0.00	1	4	12	1.550	6.700	2.000
17	0.00	0.00	0.00	1	4	12	1.575	6.125	1.950

**Table 4 materials-16-05716-t004:** Model analysis of variance (ANOVA) results for penetration.

Source	Sum of Squares	df	Mean Square	*F* Value	*p*-Value *P* Rob > *F*	Contribution (%)	
Model	0.23	9	0.026	5.21	0.0203		significant
A-DP-Amplitude	0.021	1	0.021	4.18	0.0802	8.281	
B-DP-Frequency	0.022	1	0.022	4.44	0.0732	8.796	
C-Travel Speed	0.056	1	0.056	11.44	0.0117	22.662	
AB	0.003025	1	0.003025	0.62	0.458	1.228	
AC	0.026	1	0.026	5.38	0.0534	10.658	
BC	0.052	1	0.052	10.55	0.0141	20.899	
A^2^	0.002342	1	0.002342	0.48	0.5118	0.951	
C^2^	0.042	1	0.042	8.57	0.0221	16.977	
A^2^B	0.024	1	0.024	4.82	0.0641	9.548	
Residual	0.034	7	0.004905				
Lack of Fit	0.016	3	0.005462	1.22	0.4118		not significant
Pure Error	0.018	4	0.004488				
Cor Total	0.26	16					
Std. Dev.	0.07	R-Squared	0.8701				
PRESS	0.33	Adeq Precision	8.625				

**Table 5 materials-16-05716-t005:** Model ANOVA results for width.

Source	Sum of Squares	df	Mean Square	*F* Value	*p*-Value *P* Rob > *F*	Contribution (%)	
Model	2.78	9	0.31	8.85	0.0045		significant
A-DP-Amplitude	0.58	1	0.58	16.58	0.0047	22.8570	
B-DP-Frequency	0.019	1	0.019	0.54	0.4853	0.7444	
C-Travel Speed	0.0006125	1	0.0006125	0.018	0.8983	0.0248	
AB	0.32	1	0.32	9.24	0.0188	12.7382	
BC	0.44	1	0.44	12.6	0.0094	17.3702	
A^2^	0.031	1	0.031	0.9	0.374	1.2407	
B^2^	0.32	1	0.32	9.05	0.0197	12.4762	
C^2^	0.28	1	0.28	7.94	0.0258	10.9460	
A^2^B	0.55	1	0.55	15.67	0.0055	21.6025	
Residual	0.24	7	0.035				
Lack of Fit	0.05	3	0.017	0.34	0.799		not significant
Pure Error	0.19	4	0.049				
Cor Total	3.02	16					
Std. Dev.	0.19	R-Squared	0.9192				
PRESS	1.46	Adeq Precision	10.302				

**Table 6 materials-16-05716-t006:** Model ANOVA results for height.

Source	Sum of Squares	df	Mean Square	*F* Value	*p*-Value *P* Rob > *F*	Contribution (%)	
Model	0.8	9	0.089	4.25	0.0347		significant
A-DP-Amplitude	0.028	1	0.028	1.32	0.288	3.367	
B-DP-Frequency	0.042	1	0.042	2.01	0.1996	5.127	
C-Travel Speed	0.11	1	0.11	5.19	0.0569	13.238	
AB	0.0007563	1	0.0007563	0.036	0.8545	0.092	
AC	0.11	1	0.11	5.45	0.0522	13.901	
BC	0.13	1	0.13	6.31	0.0403	16.094	
A^2^	0.16	1	0.16	7.68	0.0276	19.589	
B^2^	0.006737	1	0.006737	0.32	0.5879	0.816	
C^2^	0.23	1	0.23	10.89	0.0131	27.776	
Residual	0.15	7	0.021				
Lack of Fit	0.04	3	0.013	0.49	0.7057		not significant
Std. Dev.	0.14	R-Squared	0.8454				
PRESS	0.8	Adeq Precision	6.952				

## Data Availability

Not applicable.

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
