# Peer review of "Process Optimization, Microstructure and Mechanical Properties of Wire Arc Additive Manufacturing of Aluminum Alloy by Using DP-GMAW Based on Response Surface Method"

_materials, 2023, doi:10.3390/ma16165716_

Round 1

Reviewer 1 Report

1.      The abstract section is poorly written. It seems like an introduction section. The authors are advised to add important results and novelty in the abstract section.

2.      Introduction Section: “The response surface method (RSM) is to fit the equation from the data of the experiment…….”- It can be started as a new paragraph.

3.      The author is advised to add more recent literature on RSM.

4.      How the process parameters were decided in Table 2.

5.      Table 3: How there are only 17 experimental datasets when you have considered 3 input parameters. Please explain the conditional assumption in the design.

6.      In equations (1) to (3), the values of P, W, and H have residual values of 1.57, 6.39, and 1.91 respectively. Can we assume that when all the input parameters A, B, and C have a zero value, these outputs will exhibit the above residual value results?

7.      These regression equations need more explanation in line with the experimental results.

8.      The ANOVA tables should contain another column for the percentage of contribution of each parameter.

9.      Table 4: The individual parameters A and B have a P-value of 8% and 7% respectively. It means these are not significant in determining the penetration. Also, there is no square term for B-square. What is A2B.

10.  Similar observations can be found in Tables 4 and 5 as well. It shows the individual terms did not significantly affect the output responses.

11.  The ANOVA results should be appropriately analyzed with respect to the experimental results. It does not mean anything by simply putting software-generated data without explanation.

12.  The manuscript should focus more on the microstructural analysis section, which needs to be substantially explained.

13.  Fig. 8 (EDS result): How the author can conclude that the Si elements are uniformly distributed. There are dendritic formations shown in the Figure.

14.  Fig. 9 (Tensile results): How many times the tensile results are repeated? Provide error bars in the graph.

Author Response

  1. The abstract section is poorly written. It seems like an introduction section. The authors are advised to add important results and novelty in the abstract section.

A: Thanks for your suggestion. The important results and innovations are emphatically described.

  1.      Introduction Section: “The response surface method (RSM) is to fit the equation from the data of the experiment…….”- It can be started as a new paragraph.

A: Thanks for your suggestion. We change the "The response surface method (RSM) is to fit the equation from the data of the experiment……." in the Introduction Section to the beginning of the new paragraph.

  1.      The author is advised to add more recent literature on RSM.

A: Thanks for your suggestion. We have added more recent literature on RSM in the introduction Section.

  1.      How the process parameters were decided in Table 2.

A: Thanks for your suggestion. The process parameters in Table 2 are set by the maximum adjustable range of the welding power source (Double pulsed frequency 0-7Hz, Double pulsed amplitude 0-2m/min).

  1.      Table 3: How there are only 17 experimental datasets when you have considered 3 input parameters. Please explain the conditional assumption in the design.

A: Thanks for your suggestion. In order to investigate the fitting of the central area and ensure the repeatability of the test, the central point repeated test is set to the maximum number of groups (5 groups). We added explanations to "2. Materials and methods".

  1.      In equations (1) to (3), the values of P, W, and H have residual values of 1.57, 6.39, and 1.91 respectively. Can we assume that when all the input parameters A, B, and C have a zero value, these outputs will exhibit the above residual value results?

A: Thanks for your suggestion. The multiple quadratic regression equation is used to fit the functional relationship between the factors (A double pulse amplitude, B double pulse frequency and C travel speed) and the response values (width W, depth P and height H). In the formula, the values of A, B and C should be within the range of parameters designed for the test.

  1.      These regression equations need more explanation in line with the experimental results.

A: Thanks for your suggestion. According to the test results, we added more explanations to the regression equation in "3. Results and discussion".

  1.      The ANOVA tables should contain another column for the percentage of contribution of each parameter.

A: Thanks for your suggestion. In Design-expert software, the percentage of contribution of each factor is not provided.

  1.      Table 4: The individual parameters A and B have a P-value of 8% and 7% respectively. It means these are not significant in determining the penetration. Also, there is no square term for B-square. What is A2B.

A: Thanks for your suggestion. In determining the penetration depth, the influence of C travel speed is important, while A and B are not significant in determining the penetration depth. For the significance of the model, we used automatic model selection (Model-Auto Select...) to optimize the model when the algorithm selects the model. We added a description to "3. Results and discussion".

  1.  Similar observations can be found in Tables 4 and 5 as well. It shows the individual terms did not significantly affect the output responses.

A: Thanks for your suggestion. According to ANOVA Table 5, A double pulse amplitude has a significant impact on the width, while B double pulse frequency and the C travel speed have no significant impact on the width. We also added a note in section 3.

  1.  The ANOVA results should be appropriately analyzed with respect to the experimental results. It does not mean anything by simply putting software-generated data without explanation.

A: Thanks for your suggestion. In "3. Results and discussion", we added appropriate analysis on the results of ANOVA and experimental results.

  1.  The manuscript should focus more on the microstructural analysis section, which needs to be substantially explained.

A: Thanks for your suggestion. In the part of microstructure analysis, we added more substantial explanations on the microstructure of DP-GMAW aluminum silicon alloy additive manufacturing.

  1.  Fig. 8 (EDS result): How the author can conclude that the Si elements are uniformly distributed. There are dendritic formations shown in the Figure.

A: Thanks for your suggestion. This is our negligence, and the description here is not very accurate. α-Al grains are mainly dendritic formations,containing only a small amount of silicon. Most silicon elements are concentrated in Al-Si eutectic. We have made corresponding modifications in the text.

  1. 9 (Tensile results): How many times the tensile results are repeated? Provide error bars in the graph.

A: Thanks for your suggestion. The transverse and vertical tensile specimens were repeated twice respectively. We added error bars in Figure 9.

Reviewer 2 Report

Dear authors, the subject of the article is current and it is very likely that it can be used in the metallurgical industry.

The introduction is written correctly. Citations are up-to-date and well-adjusted to the issues raised.

In point 2. Materials and methods, the authors should describe in detail the methodology of moving the welding head and specify the method of determining its speed.

Figure one should include the transfer head.

In point 3, the "x" means vector multiplication, please replace it with a dot "∙"

In Figure 6, please mark the approximate places from which the metallographic specimens were made.

In the case of photos of crack fractures, please expand the analysis with macrofractographic photos and describe the cracking mechanism in detail.

Conclusions - correct

Author Response

In point 2. Materials and methods, the authors should describe in detail the methodology of moving the welding head and specify the method of determining its speed.

A: Thanks for your suggestion. In point 2. Materials and methods, we have made a detailed supplement to the control method and determination method of travel speed.

Figure one should include the transfer head.

A: Thanks for your suggestion. We added a motion control system to Figure 1.

In point 3, the "x" means vector multiplication, please replace it with a dot "∙"

A: Thanks for your suggestion. We will change the "x" in point 3 to "∙".

In Figure 6, please mark the approximate places from which the metallographic specimens were made.

A: Thanks for your suggestion. In Figure 6, we added the general position of metallographic specimen.

In the case of photos of crack fractures, please expand the analysis with macrofractographic photos and describe the cracking mechanism in detail.

A: Thanks for your suggestion. Our statement is not very accurate. We deleted the description of macro cracks, because no macro cracks were found in the metallographic samples.

Reviewer 3 Report

The paper entitled "Process optimization, microstructure and mechanical properties of wire arc additive manufacturing of aluminum alloy by using DP-GMAW based on response surface method" presents a single-pass deposition layer parameters (double pulse amplitude, double pulse frequency and travel speed) of DP-GMAW with optimization using the response surface method (RSM).  From my point of view, the topic is of great interest but the overall quality should be improve:

·        A graphical abstract would add interest to catch the eye

·        In the introduction please end with the novelty of your work not with reference to prior works

·        Add reference to recent works on the same field:

o   https://doi.org/10.1016/j.jmatprotec.2021.117271

o   https://doi.org/10.1016/j.jmrt.2021.04.076

o   https://doi.org/10.1016/j.measurement.2021.110452

·        References are not correctly address in the text

·        Name the materials following the standart

·        What colour means in Figure 3

·        Have you use topological optimization?

·        Analize wall width in Figure 6 c) (not correctly named (a))

·        Scale is missing in Figure 7

·        It is possible to have the curve of strain stress

·        Yield Strength result is not shown

·        The conclusions could be enriched and presented in bullet format.

These comments are intended to improve a correct experimental works done in AM

Author Response

      A graphical abstract would add interest to catch the eye

A: Thanks for your suggestion. 

  •        In the introduction please end with the novelty of your work not with reference to prior works

A: Thanks for your suggestion. We modified the description of the last paragraph of the introduction and added a description of the work and novelty of this article.

  •        Add reference to recent works on the same field:

A: Thanks for your suggestion. We have added references to recent works in the same field.

o   https://doi.org/10.1016/j.jmatprotec.2021.117271

o   https://doi.org/10.1016/j.jmrt.2021.04.076

o   https://doi.org/10.1016/j.measurement.2021.110452

  •        References are not correctly address in the text

A: Thanks for your suggestion. 

  •        Name the materials following the standart

A: Thanks for your suggestion. We modified the material naming according to the standard.

  •        What colour means in Figure 3

A: Thanks for your suggestion. The colors in Figure 3 represent responder values of different sizes. We have supplemented the explanation in the text and the figure.

  •        Have you use topological optimization?

A: Thanks for your suggestion. We do not use topology optimization.

  •        Analize wall width in Figure 6 c) (not correctly named (a))

A: Thanks for your suggestion. We modified the naming of Figure 6c.

  •        Scale is missing in Figure 7

A: Thanks for your suggestion. We added a scale bar to Figure 7.

  •        It is possible to have the curve of strain stress

A: Thanks for your suggestion. We made statistics on the results of tensile tests, and no stress-strain curve was generated.

  •        Yield Strength result is not shown

A: Thanks for your suggestion. We analyzed its properties according to the ultimate tensile strength and elongation, and did not count the yield strength.

  •        The conclusions could be enriched and presented in bullet format.

A: Thanks for your suggestion. We have adjusted the conclusion and presented it in bullet format to enrich it.

These comments are intended to improve a correct experimental works done in AM

Round 2

Reviewer 1 Report

  1. The ANOVA tables should contain another column for the percentage of contribution of each parameter. The author replied that In Design-expert software, the percentage of contribution of each factor is not provided. The author can manually calculate and provide the column. 

Author Response

Thanks for your suggestion. We added a column of contribution percentage of each factor in the ANOVA table.

Reviewer 2 Report

Dear authors,

I accept the changes made.

I think the article is publishable.

Author Response

Thank you for your recommendation

Reviewer 3 Report

It has improved

Author Response

Thank you for your recommendation